# Longitudinal Resilience and Risk Factors in Pediatric Postoperative Pain (LORRIS): Protocol for a Prospective Longitudinal Swiss University Children's Hospitals-Based Study

Jana Hochreuter,[1,2,3] Thomas Dreher,[4,5] Carol-Claudius Hasler,[6] Sandro Canonica,[4,5] Cosima Locher [7,8] Ulrike Held [9] Jennifer Rabbitts,[10] Helen Koechlin [1,2,3]

For numbered affiliations see end of article.

**Correspondence to**
Dr Helen Koechlin;
helen.koechlin@kispi.uzh.ch

## ABSTRACT

**Introduction** Chronic postsurgical pain (CPSP) is defined as pain that persists after a surgical procedure and has a significant impact on quality of life. Previous studies show the importance of psychological factors in CPSP, yet the majority of studies focused solely on negative emotions. This longitudinal observational study aims to broaden this knowledge base by examining the role of emotional state, emotion variability, emotion regulation and emotion differentiation on the child and the parent level for the development CPSP, and to describe pain and emotion-related trajectories following surgery.

**Methods and analysis** We intend to include 280 children and adolescents aged 8–18 years with a planned orthopaedic surgery and their parents. A total of five assessment time points is planned: 3 weeks before surgery (baseline), 2 weeks after surgery (post) and 3 months (follow-up (FU) 1), 6 months and 12 months after surgery. At baseline and post only, children and parents are asked to complete a daily diary thrice a day for a week where they rate their current emotional state and their pain severity (children only). Emotional state ratings will be used to calculate indices of emotion variability, emotion regulation and emotion differentiation. Children and parents will complete questionnaires at each time point, including measures on quality of life, social support, sleep, and symptoms of anxiety and depression.

To predict development of CPSP, generalised linear regression models will be used, resulting in ORs and 95% CIs. Pearson product-moment correlations between predictors and outcomes will be evaluated at each time point. The primary outcome of the prediction model is CPSP at FU1. For the trajectory analysis, the classification method K-means for longitudinal data will be used to determine clusters in the data.

**Ethics and dissemination** The Ethics Committee of the Canton of Zurich, Switzerland, has approved the study (ID: 2023-01475). Participants will be compensated, and a dissemination workshop will be held.

**Trial registration number** NCT05816174.

## STRENGTHS AND LIMITATIONS OF THIS STUDY

⇒ This is a prospective, multicentre, longitudinal hospital-based study and the sample size allows to examine a range of risk and resilience factors.
⇒ The longitudinal study design permits measurement of pain-related and emotion-related trajectories up until 12 months after surgery.
⇒ The repeated assessments design will lead to an in-depth understanding of changes of risk and resilience factors over time.
⇒ The primary limitation of this study is the potential loss to follow-up and missing datapoints due to several points of measurement.
⇒ The results will be mainly based on self-reports, which are considered gold standard but can be subject to bias.

## INTRODUCTION

Chronic postsurgical pain (CPSP) is defined as chronic pain that persists over time (at least 3 months) following a surgical procedure and that has a significant impact on physical and psychosocial quality of life.[1 2] Meta-analytical estimates in children and adolescents describe that around 25% of paediatric patients undergoing major surgery report chronic pain 3–12 months after surgery.[2] Highest pain intensity ratings are observed in 2 weeks following surgery, with a steady decrease afterwards over the following 6 months,[3] and pain ratings 2 weeks after surgery have been shown to predict the development of moderate to severe pain 1 year later.[4 5] Further, most children and adolescents experience decreases in their physical activity levels following surgery, and it has been suggested that objectively monitored physical activity levels could be useful to indirectly assess functional recovery.[6] A recent study in adolescents undergoing spinal fusion

found that a majority of participants were characterised by a steep decrease of physical activity after surgery.[7]

Previous research has identified psychosocial risk factors that are associated with the development of CPSP, among them presurgical pain intensity, child anxiety, child pain coping efficacy, child sleep patterns and parental pain catastrophising.[2 8 9] Generally, emotions and emotion-related factors are known to influence pain, pain-related disability and pain-related distress.[10 11] There is some research on the role of emotion regulation in chronic pain,[12] and the role of negative emotions is fairly well documented in the pain experience.[10 13 14] However, other emotion-related factors such as emotion differentiation and emotion variability have yet to be studied in the context of paediatric CPSP, as they present as good targets for prevention and treatment modules and can be screened for in clinical practice.

### Emotion variability

Emotion variability describes the range of emotional fluctuations around an individual's average emotional intensity.[15 16] Greater emotion variability has been linked to more mental health symptoms in adolescents[17 18] and higher emotion regulation demands.[19] In paediatric chronic pain, one study found greater positive emotion variability to be associated with less pain-related interference and more engagement in activities despite pain.[16] Emotion variability can be measured using the within-person SD of repeated assessments of current emotional state.[16 18 20 21]

### Emotion differentiation

Emotion differentiation defines the level of specificity people use when identifying their emotional experiences.[22–24] Higher emotion differentiation indicates greater ability to differentiate between one's own emotional states on a more fine-grained level and by using precise terms.[24] High or low emotion differentiation can be calculated by means of intraclass correlations, where high consistency in emotion ratings across a predefined episode (eg, a week-long daily diary phase) suggests poorer differentiation.[24 25] Previous research has demonstrated that higher emotion differentiation is associated with better mental health (see Seah and Coifman[26] for a meta-analysis). To date, emotion differentiation has not been studied in the context of paediatric chronic pain.

### Emotion regulation

Emotion regulation refers to a person's ability to influence what emotions they have, when they have them, and how they experience and express these emotions.[27 28] A handful of previous studies has explored the role of emotion regulation in paediatric chronic pain[16 20 29 30] yielding mixed results, while some found that different types of emotion regulation, namely expressive suppression and cognitive reappraisal, were associated with more frequent hospitalisation due to pain,[29] and lower engagement in activities despite the pain,[16] others did not identify significant associations between emotion regulation and pain intensity or pain-related disability.[20 30] The habitual use of specific emotion regulation strategies is considered stable within individuals[31–33] and can be measured by means of questionnaires.[19]

### Familial influences on paediatric pain

Previous research has shown that around 50% of children and adolescents with chronic pain have a parent who has had chronic pain in the past or is currently living with it.[34] Parents and the family environment have been identified as a 'key context' (Simons *et al*[35] p. 702) regarding the understanding, assessment, and management of paediatric pain,[35] and a bidirectional relationship between child functioning and parent factors is suggested.[35–37] Regarding emotion-related factors, the family context is considered crucial: children learn the appropriateness of valence, duration and intensity of emotions and emotion expression by observing their parents and parents' emotion regulation behaviours.[38 39]

In sum, previous studies have identified several risk factors for the development of CPSP, among them symptoms of anxiety, preoperative pain intensity and poorer sleep quality as well as parental catastrophising, even though the results were equivocal in part.[5 6 40] The extent to which emotion-related factors can serve as resilience factors has had minimal investigation in the transition from acute to CPSP.[5] Further, the promotion of resilience mechanisms as a prevention strategy has had limited focus in the context of paediatric chronic pain.[7 41]

### Objectives

The aim of this observational longitudinal study is to specifically examine the potential role of emotion-related factors (namely emotion variability, emotion differentiation and emotion regulation), as well as familial influences in the development of CPSP and describe pain-related and emotion-related trajectories following major surgery up to 12 months after surgery. We aim to identify whether and which emotion-related factors increase or decrease the risk for the development of CPSP after planned orthopaedic surgery. To the best of our knowledge, this is the first study to specifically examine these emotion-related factors as both potential risk and resilience factors in the development of CPSP.

## METHODS AND ANALYSIS
### Study design and setting

This is a prospective longitudinal observational study in patients undergoing scheduled orthopaedic surgery and their parents. The study starts in January 2024 and will continue to run through the end of January 2027. As scheduled orthopaedic surgeries are planned well in advance, patients can be approached early and asked about their interest in participating in the study. See figure 1 for an overview of assessment time points. The study is registered at ClinicalTrials.gov (NCT05816174).

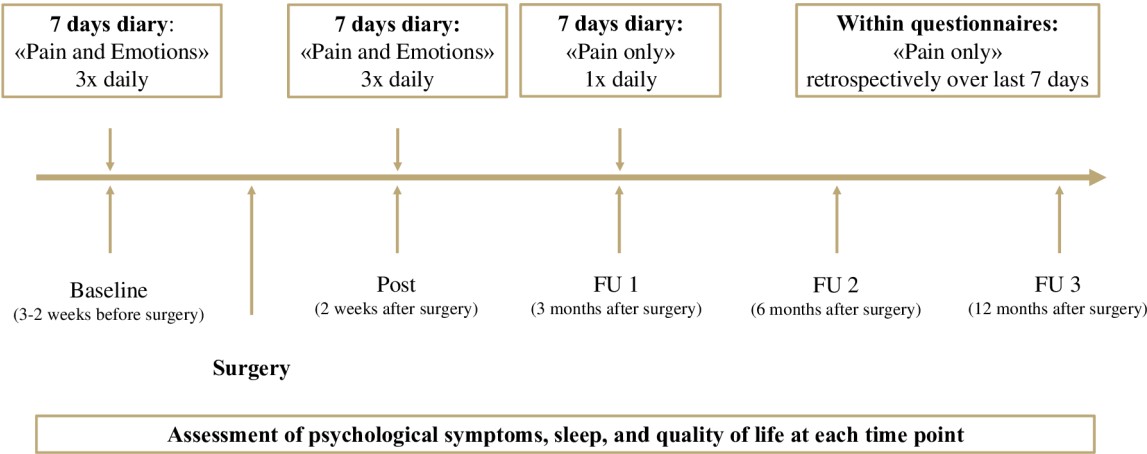

**Figure 1** Overview of assessment time points. FU, follow-up.

All participants will be recruited from three University Children's Hospitals in Switzerland.

## Patient and public involvement

Patients were not involved in the design of the study, however, the first 10 families that complete the postassessment will be interviewed and asked about their experience in the study so far. This feedback will be used to adapt study procedures if needed. The results will be disseminated to study participants via a newsletter and a dissemination event where participants, clinical collaborators and the interested public will be invited to discuss the study results.

## Sample and sample size calculation

The anticipated prevalence rate of CPSP is 25%, based on international estimates.[2 42] The aim of this study is to recruit a sample of 280 child–parent dyads, resulting in an anticipated 70 children and adolescents with CPSP. In a power calculation for a logistic regression model with 10 events per variable (EPV)), this allows to examine 7 predictor variables with CPSP status as the primary outcome. Assumptions regarding EPV will be re-evaluated following recently proposed recommendations.[43 44]

## Eligibility criteria

We will include participants between 8 and 18 years who have a planned in-patient orthopaedic surgery at one of the study centres. The child–parent dyads must be able to read (for younger children: with help) and understand German. Exclusion criteria will be serious comorbid health condition (eg, cancer, severe neurological impairment, chronic illness requiring daily medication) and prior major surgery (eg, prior spine surgery). Mental health comorbidities are no reason for exclusion.

## Procedures

Potential participants will be informed about the study by the clinical collaborators at each study site during one of their presurgical visits (eg, anaesthesia preparation interview, scheduling visit). Based on the clinical routine of the study centres, the procedures might differ slightly. If patients fall within the required age range, the clinical collaborators will provide the family with a leaflet with the most important study information. On the leaflet, a QR-Code brings families directly to a site on the website of the University Children's Hospital Zurich (main study centre), where they can find the most important information again. If families express their initial interest about the study, the physician will take a note and inform the study team, which will then contact the family. A member of the study team will then approach the respective family at their next visit at the hospital or via phone call and go through the informed consent/assent procedure with them. In all cases, parents sign the consent form for their own participation. In children under the age of 14 years, parents also sign the consent form for their child's participation, while adolescents aged 14 years or more sign their own consent form. This is in line with the Human Research Act,[45] which states that adolescents aged 14 years and more who are capable of judgement and who are to participate in a study with minimal risks can sign their own consent form. In participating hospitals, patients typically have (at least) two visits before the surgery. Once the written informed consent and assent is given, families will complete a battery of questionnaires as baseline measures before the surgery via direct link to the online questionnaires (see next section and table 1 for an overview; see online supplemental file 1 for an overview of reliability, validity, benefits and limitations of all questionnaires), using Research Electronic Data Capture (RedCap[46]). In addition, participants will be asked to download the smartphone application SEMA V.3,[47] a free tool for ecological momentary assessment provided by the University of Melbourne, Australia, especially developed for research needs.

Families will be called again 2 weeks after the surgery and asked to complete the postassessment via direct link to the questionnaires. Families will be contacted again 3 months (follow-up (FU) 1), 6 months (FU 2) and 12 months (FU 3) after surgery to complete the FU surveys and asked to complete the surveys at each assessment point within 1 week after receiving the link to the survey;

**Table 1** Measures, tools, domain and time points for data collection

| Domain | Measure | | Completed by | BL | Post | FU 1 | FU 2 | FU 3 |
|---|---|---|---|---|---|---|---|---|
| Demographics | Age, gender, school, school attendance and performance; family constellation, parental educational level and work status | Parent and self-report: online baseline questionnaire | Parent(s) and child | X | | | | |
| Medical history | Neonatal intensive care (yes/no), previous surgeries (yes/no; which one), child's pain history (pain duration, pain location) | Parent report: online baseline questionnaire | Parents | X | | | | |
| | Complications in the hospital (yes/no) Pain treatment immediately following surgery | Child's medication list in electronic health record | Data entered by study team | | X | | | |
| Experience-sampling method | Positive and negative emotions (emotion variability and emotion differentiation)* | Items from the Positive and Negative Affect Schedule[48] | Daily diary: child and parent | X | X | | | |
| | Pain severity | Numeric Rating Scale (NRS), 0–10 | Daily diary: child | X | X | X | | |
| Psychosocial factors child | | | | | | | | |
| | Pain severity (intensity, distress, interference[77])* | Numeric Rating Scale (NRS), 0–10 | Child self-report | | | | X | X |
| | Preoperative screening | Paediatric Pain Screening Tool[51 52] | Child self-report | X | | | | |
| | Functional disability | Functional Disability Index[54] | Child self-report | X | X | X | X | X |
| | Fear of pain | Fear of Pain Questionnaire[53] | Child self-report | X | | | | |
| | Pain catastrophising | Pain Catastrophising Scale—three-item version[78] | Child self-report | X | | | | |
| | Attachment; parent–child relationship | Security Scale[79] | Child self-report | X | | | | |
| | Symptoms of anxiety and depression* | Revised Child Anxiety and Depression Scale-short version[56] | Child self-report | X | X | X | X | X |
| | Sleep | Pittsburgh Sleep Quality Index[57] | Child self-report | X | X | X | X | X |
| | Sensory processing sensitivity | Highly Sensitive Child Scale[58] | Child self-report | X | | | | |

Continued

**Table 1** Continued

| Domain | | Measure | Completed by | BL | Post | FU 1 | FU 2 | FU 3 |
|---|---|---|---|---|---|---|---|---|
| | Emotion regulation* | Emotion Regulation Questionnaire—Child and Adolescent version[19] | Child self-report | X | | | | |
| | Quality of life | Paediatric Quality of Life Inventory[59] | Child self-report | X | X | X | X | X |
| | Social support | Social Support Questionnaire for Children[60] | Child self-report | X | | | | |
| | Post-traumatic stress symptoms | Child and Adolescent Trauma Screening[80] | Child self-report | X | | X | | |
| Psychosocial factors parents | | | | | | | | |
| | Parents' own pain history | 2 questions: '(1) Have you ever and (2) are you currently living with persistent or recurrent pain for at least 3 months?' | Parent self-report | X | | | | |
| | Postoperative pain measure (child's pain) | Postoperative Pain Measure for Parents[62] | Parent report | | X | | | |
| | Fear of pain | Parent Fear of Pain Questionnaire[35] | Parent self-report | X | | | | |
| | Pain catastrophising | Pain Catastrophising Scale—three-item version[78] | Parent self-report | X | | | | |
| | Anxiety and depression* | Anxiety and Depression Scale[63] | Parent self-report | X | X | X | X | X |
| | Sensory processing sensitivity | Highly Sensitive Person Scale[65] | Parent self-report | X | | | | |
| | Emotion regulation* | Emotion Regulation Questionnaire[31] | Parent self-report | X | | | | |
| | Quality of life | WHO Well-Being Index[66] | Parent self-report | X | X | X | X | X |
| Primary outcome | | | | | | | | |
| Chronic postsurgical pain | | Presence of CPSP: pain diary for seven consecutive days (NRS)+health-related quality of life (PedsQL, see above) | | | | X | | |

*Primary predictors are marked with a *. Unmarked variables are considered secondary or exploratory.
BL, baseline; CPSP, chronic postsurgical pain; FU, follow-up; PedsQL, Paediatric Quality of Life Inventory.

up to 3-weekly reminders will be sent after the initial message.

In addition to the questionnaires at each assessment point, all participants (ie, children and parents) will complete daily diaries: To estimate the influence of emotion-related factors on pain trajectories, children and adolescents will complete two daily diary phases (ie, 2 weeks of daily diary) during which they rate both their pain severity and emotional state (referred to as 'pain and emotions' throughout the protocol; at baseline and postassessment), and one daily diary episode during which they only rate their pain intensity (referred to as 'pain only' throughout the protocol; at FU 1, see figure 1). Parents will complete two daily diary phases (at baseline and post) during which they exclusively rate their emotional state.

Daily diary—pain and emotions (two time points): At baseline and postassessment, children and adolescents will complete ratings of their pain severity on an 11-point Numeric Rating Scale with anchors 0='no pain' and 10='worst pain possible'. Pain severity includes pain intensity, pain-related distress and pain-related interference; hence, participants will give three different ratings.[41 42] The pain ratings will be used to classify the presence of acute postsurgical pain at the postassessment (in combination with a health-related quality of life rating; see overview of questionnaires for this time point in table 1). In addition, all participants will rate the extent to which they experience positive and negative emotions three times each day on their smartphone and during their normal daily routine over the course of 7 days. These ratings will be used to calculate a score of emotion differentiation (ie, the ability to differentiate between emotional states), a mean value of positive and negative emotions and an index of emotional variability[26 36 37]. The items of the short version of the Positive and Negative Affect Schedule for Children (PANAS-C) will be used to assess emotional state.[38–40]

Daily diary—pain only (once): To diagnose CPSP, participants will be asked to rate their daily pain severity for 7 days starting at FU1. Participants will be prompted at the end of each day to respond on an 11-point numeric rating scale with anchors 0='no pain' and 10='worst pain possible'. In addition, daily medication use will be indicated (yes/no and listing medication in a free text field).

Number of steps taken each day will be counted as an indicator of changes in physical activity before and after surgery (ie, at the postassessment), measured by means of accelerometer (GENEActive, Activeinsights), an ambulatory, non-invasive activity-monitoring device that registers physical activity and can be worn around the wrist.

## Study measures
See table 1 for an overview, see online supplemental file 1 for an overview of reliability, validity, benefits and limitations of all questionnaires. For all measures, we use the validated German version (if available) or translate and back-translate the items with the help of a professional interpreter or native speaker. Measures of primary interest are marked with a * in table 1, all other variables are considered exploratory.

## Child measures
Emotional state: The PANAS-C[48–50] is a measure of positive and negative affect by providing mood adjectives (eg, 'scared', 'happy'). Participants are asked to rate 10 items on a 5-point Likert scale from 1 (=very slightly or not at all) to 5 (=very much) to assess the extent to which they currently feel each of five positively and five negatively valenced emotions.

Pain screening tool: The Paediatric Pain Screening Tool[51 52] entails nine self-reported items that belong to a physical and psychosocial subscale. Answers are given dichotomously (disagree=0, agree=1). Higher scores indicate higher risk status of poor pain-related outcomes.

Fear of pain: The Fear of Pain Questionnaire for Children (FOPQ-C)[53] is a 24-item self-report inventory to assess fear of pain. Participants are asked to rate each item on a Likert scale from 0 (=strongly disagree) to 4 (=strongly agree). A higher total score indicates higher fear.

Functional disability: Children's self-reported difficulty in physical and psychosocial functioning due to their physical health is assessed by the Functional Disability Inventory.[54] 15 items are rated on a Likert scale from 0 (=no problems) to 4 (=not possible) and concern perceptions of activity limitations during the past 2 weeks. Higher scores indicate greater disability.

Security in parent–child relationships: The Security Scale[55] consists of 15 items and assesses participants' perceptions of security in parent–child relationships. Items are presented in a format of statements, participants choose which statement is more characteristic of them and then rate the statement on a Likert scale from 1 (=really true) to 4 (=sort of true). Higher scores indicate a more secure attachment.

Symptoms of anxiety and depression: The Revised Child Anxiety and Depression Scale-short version (RCADS-short version)[56] is a 25-item self-report questionnaire used to assess symptoms of depression and anxiety. Participants are asked to indicate how often each item applies to them according to a 4-point Likert scale from 0 (=never) to 3 (=always). High scores indicate high anxiety and depression symptoms.

Sleep: The Pittsburgh Sleep Quality Index[57] is the most commonly used retrospective self-report questionnaire that measures sleep quality over the previous month. We will use a version that has been slightly adapted for children by researchers of the University Children's Hospital Zurich. This version contains of 25 items, and each item is rated on a scale from 0 (=no difficulty) to 3 (=severe difficulty). In addition to a global sleep quality factor, seven domains of sleep difficulties can be assessed: Sleep quality, sleep latency, sleep duration, habitual sleep efficiency, sleep disturbances, use of sleeping medications and daytime dysfunction. A higher global score indicates more sleep difficulties.

Sensory processing sensitivity: The Highly Sensitive Child Scale (HSC)[58] is a 12-item self-reported questionnaire. Participants are asked to rate each item on a Likert scale from 1 (=not at all) to 7 (=extremely). In addition to a total score, three subscales can be calculated: low sensory threshold (LST; three items, eg, 'I don't like watching TV programmes that have a lot of violence in them'), Ease of Excitation (EOE; five items, eg, 'I get nervous when I have to do a lot in little time') and aesthetic sensitivity (AES; four items, eg, 'I love nice smells'). Higher scores indicate higher sensitivity.

Emotion regulation: The Emotion Regulation Questionnaire for Children and Adolescents (ERQ)[19] is a self-report questionnaire containing 10 items. Items are

rated on a Likert scale from 1 (=strongly disagree) to 7 (=strongly agree), and assess individual differences in the habitual use of two emotion regulation strategies, namely cognitive reappraisal (eg, 'When I'm faced with a stressful situation, I make myself think about it in a way that helps me stay calm') and expressive suppression (eg, 'I keep my emotions to myself'). Higher scores indicate more habitual use of the respective emotion regulation strategy.

Quality of life: The Paediatric Quality of Life Inventory (PedsQL)[59] is a self-or parent-reported questionnaire for children and adolescents. 23 items are rated on a Likert scale from 0 (=never a problem) to 4 (=almost always a problem). The questionnaire consists of the subscales physical functioning (eg, 'It is hard for me to run'), emotional functioning (eg, 'I feel sad or blue'), social functioning (eg, 'It is hard to keep up with my peers') and school functioning (eg, 'It is hard to pay attention in class') in the past month. Higher scores indicate better quality of life.

Social support: The Social Support Questionnaire for Children[60] consists of 50 items. It measures children's social support across five distinct sources of support: parents, relatives, non-relative adults, siblings and peers. Items are rated on a 4-point Likert scale ranging from 0 (=never or rarely true) to 3 (=often or always true). Higher scores indicate higher levels of perceived support.

Symptoms of post-traumatic stress: The Child and Adolescent Trauma Screen 2[61] contains a checklist of 15 potentially traumatic events (PTEs), followed by an item asking which of the PTEs bothers them most. Post-traumatic stress symptoms are measured by 20 items that are rated on a 4-point Likert scale ranging from 0 (=never) to 3 (=almost always). Five yes/no items assess whether the previously rated post-traumatic stress symptoms interfere with five key areas of functioning and are used to assess psychosocial functioning.

### Parent measures

Postoperative pain: With the Postoperative Pain Measure for Parents (PPMP),[62] parents can assess and estimate their child's postoperative pain. The checklist includes 15 items where the parents select 'yes' (=1 point) or 'no' (=0 points) as to whether the child exhibits pain-related behaviour, with higher scores indicating more postoperative pain.

Fear of Pain Questionnaire: The Parent FOPQ[35] is a 23-item self-report inventory to reflect a parent's own fear associated with their child's pain experience. Parents are asked to rate each item on a Likert scale from 0 (=strongly disagree) to 4 (=strongly agree). A higher total score indicates higher fear of pain.

Symptoms of anxiety and depression: The Hospital Anxiety and Depression Scale[63] is a widely used measure of psychological distress, and indicates the presence of symptoms of depression (seven items) and anxiety (seven items) over a 1-week period. The 14 items are rated on a 4-point Likert scale ranging from 0 to 3, with higher scores indicate greater symptom severity.

Sensory processing sensitivity: The Highly Sensitive Person Scale[64 65] is a 12-item self-reported questionnaire, and participants are asked to rate each item on a Likert scale from 1 (=strongly disagree) to 7 (=strongly agree). In addition to a total score, three subscales can be calculated, as for the HSC Scale: Low Sensory Threshold, Ease of Excitation and Aesthetic Sensitivity. Higher scores reflect higher sensitivity.

Emotion regulation: The ERQ[31] assesses the habitual use of two emotion regulation strategies, namely cognitive reappraisal (six items) and expressive suppression (four items). The 10 self-reported items are rated on a 7-point Likert scale ranging from 1 (=strongly disagree) to 7 (=strongly agree). Higher scores indicate higher usage of the respective strategy.

Well-being: Parental well-being will be measured using he WHO-5 index.[66] It consists of five items that are rated on a 6-point Likert scale ranging from 0 (=never) to 5 (=always), indicating how often participants have felt in 'good spirits', 'active', 'relaxed', etc over the course of the last 2 weeks. Higher scores indicate more well-being.

### Primary outcome

The primary endpoint is CPSP at FU 1. CPSP is defined as greater than minimal pain (pain intensity ≥3) on more than 50% of days as measured by daily diary over 7 days, and impairment in health-related quality of life (score of ≤74.9 on the PedsQL,[59] ie, 1 SD below the population mean based on normative US data). The combination of pain and quality of life measures will result in a binary variable (CPSP yes/no). This measurement of CPSP is supported by previous research,[6 67 68] and a modified definition of CPSP by the International Association for the Study of Pain.[69]

### Secondary outcomes

Secondary endpoints are trajectories of pain and emotion-related factors over time (ie, baseline through FU3), number of steps taken by patients, pain severity and quality of life. Postoperative complications will be monitored using the modified Clavien-Dindo Classification.[70]

### Statistical methods

Descriptive statistics will include mean and SD or median and IQR for normally distributed and non-normally distributed or ordinal variables measured at baseline. Categorical variables will be reported as number and percentage of total. The primary outcome is binary, therefore, generalised linear regression models or generalised linear mixed effects models will be used, using a logit link function. Fixed effects will be estimated for the primary predictors (as marked in table 1). Random effects will be included in the longitudinal model to account for repeated observations in the individual participants. The resulting estimands will be ORs and 95% CIs. Variables at baseline will be used in the prediction model, with performance measures area under the curve, calibration and scaled Brier Score.[71 72] Variables

measured consecutively over multiple days at each assessment interval will be averaged over time. We will use the rule of 10 EPV that would determine the number of independent variables in the regression model. Potentially time-varying covariates will be addressed. Potentially non-linear relationships of the independent variables with outcome will be evaluated. Further, the purpose of the multivariable regression analysis is to develop a risk score for CPSP. This will allow to identify variables that increase or decrease the risk for CPSP, by combining them into a multivariable model.

The preselected variables (ie, the primary predictors) will be assessed for collinearity with pair-wise scatter plots, pair-wise interaction terms will be included successively into the model. For the preselected variables, no variable selection will be performed. In an exploratory analysis, potential other variables will be evaluated and the process will be described in more detail in the statistical analysis plan.

Any missing values in the independent variables or the outcome will be addressed with 100-fold multiple imputation, using chained equations. Resulting estimates will be pooled using Rubin's rule.[73 74]

We will conduct bivariate analyses to compare study completers vs study dropouts to detect potential differences regarding participant demographic between the groups. Baseline daily pain ratings will be averaged for each participant to calculate a mean baseline pain intensity score. Postassessment and FU 1 daily pain ratings will be combined with health-related quality of life data to define binary outcome variables for acute postsurgical pain and CPSP. Acute postsurgical pain and CPSP will be defined independently.

For the trajectory analysis, the classification method K-means for longitudinal data will be used to determine clusters in the data (R package kml).[75] The analysis of clusters joint trajectories (R package kml3d)[75] will be applied, as this analysis allows summarising several correlated continuous variables into a single nominal variable.[76] The trajectory analyses will be considered exploratory. Alternative clustering methods would be based on the risk score and participants with high risk for developing CPSP can be identified based on a cut-off of 10% predicted probability.

A detailed statistical analysis plan will be written up and finalised before data export for analysis. All analyses will be performed with the R programming language in a fully scripted way and dynamic reporting will be used to guarantee highest level of reproducibility.

## Monitoring

The study will be monitored for quality and regulatory adherence. The monitor, who is not involved in our study and supervised by the Clinical Trials Centre of the University Hospital Zurich, Switzerland, will ensure that the specifications of the project plan and the associated processes are adhered to.

## ETHICS AND DISSEMINATION

The Ethics Committee of the Canton of Zurich, Switzerland, has approved the study and all study-related procedures including the consent procedures for minors under the age of 18 years (ID: 2023-01475). All participants will be compensated for their time by means of a voucher. We will reimburse participating families per assessment time point, so that families who drop out will still receive a certain amount of compensation. The first few participating families will be asked about general accessibility of the study. Their feedback will be implemented in study procedures if necessary.

The risk of adverse effects due to the study is considered to be minimal. All project data will be handled with the uttermost discretion and will only be accessible to the study team. Data will be stored on a secured and password-protected server.

This study is registered at ClinicalTrials.gov (identification number: NCT05816174). After study completion, the study team organises a dissemination workshop aiming at youth with CPSP and their parents, researchers, clinicians and the interested public, to discuss study results, clinical implications, and formulate future research questions.

**Author affiliations**
[1]Department of Psychosomatics and Psychiatry, University Children's Hospital, University of Zurich, Zurich, Switzerland
[2]Division of Child and Adolescent Health Psychology, Department of Psychology, University of Zurich, Zurich, Switzerland
[3]Children's Research Centre University Children's Hospital Zurich, University of Zurich, Zurich, Switzerland
[4]Department of Pediatric Orthopedics and Traumatology, University Children's Hospital, Zurich, Switzerland
[5]Department of Pediatric Orthopedics, Orthopedic University Hospital Balgrist, Zurich, Switzerland
[6]Department of Orthopedics, University Children's Hospital Basel, Basel, Switzerland
[7]Department of Consultation-Liaison Psychiatry and Psychosomatic Medicine, University Hospital Zurich, Zurich, Switzerland
[8]Faculty of Health, Plymouth University, Plymouth, UK
[9]Department of Biostatistics and Epidemiology, Biostatistics and Prevention Institute, University of Zurich, Zurich, Switzerland
[10]Anesthesiology, Perioperative and Pain Medicine, Pediatric Anesthesiology, Stanford University, Stanford, California, USA

**Contributors** HK has designed the study, obtained funding, wrote the first draft of the manuscript and acts as project leader and sponsor for the study described in this protocol. JH has collaborated closely with HK in the process of detailing the study plan, has written and revised the manuscript and is the PhD student in this project. C-CH, TD and SC are the clinical collaborators of this study. They have all provided substantial input in the research plan and the study procedures, and have critically revised the manuscript. JR has provided substantial input and feedback on the research plan, choice of outcome variables and study procedures, and has critically revised the manuscript. CL has provided substantial feedback on the design of the study and all study related procedures, and has critically revised the manuscript. UH is responsible for data analysis in this project, has written the data analysis plan, provided advice on the power calculation and has critically revised the manuscript. All authors approved the final version of the manuscript and agreed to be accountable for all aspects of the work in ensuring that questions related to the accuracy or integrity of any part of the work are appropriately investigated and resolved.

**Funding** This work is supported by a Swiss National Science Foundation grant awarded to HK: PZ00P1_208850 (Title: 'Understanding and preventing chronic

postsurgical pain in children'). JR is supported by National Institutes of Health National Institutes of Arthritis, Musculoskeletal and Skin Diseases under award number by K24AR080786 (PI: JR).

**Disclaimer** The content is solely the responsibility of the authors and does not necessarily represent the official views of the National Institutes of Health.

**Competing interests** None declared.

**Patient and public involvement** Patients and/or the public were not involved in the design, or conduct, or reporting, or dissemination plans of this research.

**Patient consent for publication** Not applicable.

**Provenance and peer review** Not commissioned; externally peer reviewed.

**Open access** This is an open access article distributed in accordance with the Creative Commons Attribution 4.0 Unported (CC BY 4.0) license, which permits others to copy, redistribute, remix, transform and build upon this work for any purpose, provided the original work is properly cited, a link to the licence is given, and indication of whether changes were made. See: https://creativecommons.org/licenses/by/4.0/.

**ORCID iDs**
Cosima Locher http://orcid.org/0000-0002-9660-0590
Ulrike Held http://orcid.org/0000-0003-3105-5840
Helen Koechlin http://orcid.org/0000-0001-6680-8027

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
