## [Reviewer comments · BMJ Open]

ARTICLE DETAILS

TITLE (PROVISIONAL)	Longitudinal Resilience and Risk Factors in Pediatric Postoperative Pain (LORRIS): Protocol for a Prospective Longitudinal Swiss University Children's Hospitals-Based Study
AUTHORS	Hochreuter, Jana; Dreher, Thomas; Hasler, Carol-Claudius; Canonica, Sandro; Locher, Cosima; Held, Ulrike; Rabbitts, Jennifer; Koechlin, Helen

VERSION 1 – REVIEW

REVIEWER	Xu, Simao Chengdu Sport University
REVIEW RETURNED	13-Oct-2023

GENERAL COMMENTS	This study is the first study to specifically examine the role of emotion-related factors on the child and the parent level in the development of chronic pain after surgery. It has certain academic and practical value. Existing problems: 1 Overall, the writing was not standardized enough; 2 The writing logic was common; 3 Lack of necessary description of research results and conclusions in the abstract; 4 The expression of the research results in the main body was too simple and cannot fully match the research indicators and statistical methods; 5 The discussion was not in-depth enough and targeted enough, and the overall level was common.
--

REVIEWER	Vanlinthout, Luc KU Leuven University Hospitals Leuven Gasthuisberg Campus Hospital Pharmacy
REVIEW RETURNED	28-Dec-2023

GENERAL COMMENTS	Title: "Longitudinal Resilience and Risk Factors in Pediatric Postoperative Pain (LORRIS): Protocol for a Prospective Longitudinal Study." Reference: bmjopen-2023-080174 Chronic pain is a serious developmental health concern that can interfere significantly with daily functioning. Children experiencing chronic pain may miss school, withdraw from social activities, and are at risk of developing internalizing symptoms in response to
--

their pain. Given these consequences, issues related to paediatric chronic pain are important to researchers and clinicians to enable the development of effective strategies to ameliorate these problems [1-3].

It is not uncommon for paediatric Chronic Postsurgical Pain (CPSP) research to face challenges in terms of study design and participant age ranges. Many studies focus on specific age groups or narrow age ranges. Due to these shortcomings, there are several gaps in our understanding of age-related differences in the prevalence of pain and the evolutionary trajectory of pain over the lifespan. In order to fully understand the prevalence of pain in children and the risk factors that predict chronification of acute postoperative pain, it is essential that pain research be more developmentally focused; that is, using broader age intervals and/or conducting more longitudinal studies of pain prevalence will increase understanding of age differences in pain prevalence. [4,5]

We would like to congratulate the authors on their protocol for a longitudinal observational study. This is intended to 1. examine the role of emotional state, emotion variability, emotion regulation, and emotion differentiation at the child and parent levels in the development of CPSP, and 2. to describe individual trajectories related to pain and emotion after surgery.

Emotion-related factors such as those measured in this study can be screened for, and they present as good targets for training modules within the prevention and treatment of pediatric CPSP.

The authors published an earlier study on the association of parental and adolescent emotion-related factors with adolescent chronic pain behaviour [6].

However some issues have to be mentioned.

The authors wrote
on page 4, lines 7-10:

“So far, most perioperative research has focused on negative emotions.”

And further

“This is the first study to specifically examine the role of emotion-related factors on the child and the parent level in the development of chronic pain after surgery”

a) Although an enormous literature exists on the psychobiology of affect, there is no singular or even preferred definition of emotion. The term is taken for granted in itself and, most often, emotion is defined with reference to a list: anger, disgust, fear, joy, sadness, and surprise. In 1981 Kleinginna and Kleinginna listed 92 different definitions of emotion, compiled from the literature on emotion.

"Emotion" is a broad and multidimensional concept that can be approached from various perspectives depending on the field of study (psychology, neuroscience, sociology, etc.) and the specific context of the research [7,8].

b) A number of child psychosocial factors have been evaluated in the prediction of the transition from acute post-operative pain to CPSP. These factors span the domains of pain-related anxiety (pain anxiety, pain catastrophizing, fear of movement), anxiety sensitivity, pain self-efficacy, chronic pain acceptance, symptoms of posttraumatic stress disorder, symptoms of depression, and general anxiety [9].

Moreover, previous studies have shown that parents' vulnerability also affects their child's pain experience and that parents' interpretation of their child's pain may affect children's pain outcomes [9].

Given this evidence from previous research, explain more clearly the novelty of the current study and how the purpose of this study differs from that of previous studies.

When selecting an instrument for the comprehensive assessment of emotional states in children, it is crucial to consider the reliability, validity and appropriateness for the age group and cultural background of the children being assessed [10]. In order to accurately interpret the outcomes, we wanted to ask the authors to include a table summarizing the reliability, validity, benefits and limitations of each of the instruments used (as part of the supplementary information).

Several studies have indicated that children from 8 years onwards can reliably self-report [11,12]. Are individual children's reading skills and their ability to reliably self-report documented prior to the start of the study?

Research highlights the importance of examining sex differences between parents and children in paediatric pain research. Therefore, the gender of the parent (mother-father) completing the questionnaire(s) and follow-up survey(s) may be significant for the outcome [13]. Can the authors comment upon this?

Statistical data-analysis

Please provide a more structured approach to data analysis, thereby following the guidelines for “Basic statistical reporting for articles published in biomedical journals.”[14]

Please:

a) Describe crystal clear the sample size calculation.

For logistic regression analysis, sample size is typically expressed in terms of events per variable (EPV), defined by the ratio of the number of events, i.e. number of observations in the smaller of the two outcome groups, relative to the number of degrees of freedom (parameters) required to represent the predictors considered in developing the prediction model. Lower EPV values in the prediction model development have frequently been associated with poorer predictive performance upon validation [15].

b) Describe the purpose of the multivariable regression analysis.

c) Identify the candidate covariables prior to conducting the multivariable regression analysis.

d) Report how any missing data will be treated in the multivariable regression analyses.

e) For either simple or multivariable regression analyses, report the statistical model, eg. for multivariable regression models with mixed effects: describe fixed and random effect terms;

f) Report how the variables will be assessed for a) collinearity and b) interaction; and describe the variable selection process by which the final model will be developed (e.g., backward-stepwise; best subset).

Trajectory analysis.

When choosing a clustering method for longitudinal data, it is essential to consider the specific characteristics of the data, such as the presence of missing values, unequal time intervals, and the underlying assumptions of the chosen method. Exploring different algorithms and tuning their parameters can help finding the most suitable method for a given data-set.

Usually the k-means clustering algorithm is used. K-means clustering for longitudinal data is a popular method for partitioning data into distinct groups based on similarity. It offers advantages in terms of not requiring prior knowledge and being easy to implement. Their limitations lie in the difficulty of determining the optimal number of clusters and assessing the accuracy of the clustering results. [16].

Did the authors also considered alternative clustering methods?
Please comment.

REFERENCES

[1] King S, Chambers CT, Huguet A, MacNevin RC, McGrath PJ, Parker L, MacDonald AJ. The epidemiology of chronic pain in children and adolescents revisited: a systematic review.

Pain. 2011 Dec;152(12):2729-38. doi: 10.1016/j.pain.2011.07.016. PMID: 22078064.

[2] McGrath PJ, Dunn-Gier J, Cunningham SJ, Brunette R, D'Astous J, Humphreys P, Latter J, Keene D. Psychological guidelines for helping children cope with chronic benign intractable pain. Clin J Pain 1986;1:229-233 PubMed .

[3] Mikkelsen M, Salminen JJ, Kautiainen H. Non-specific musculoskeletal pain in preadolescents. Prevalence and 1-year persistence. Pain 1997;73:29-35 PubMed .

[4] King S, Chambers CT, Huguet A, et al. The epidemiology of chronic pain in children and adolescents revisited: a systematic review. Pain 2011;152:2729–38 PubMed

[5] Tutelman PR, Langley CL, Chambers CT, Parker JA, Finley GA, Chapman D, Jones GT, Macfarlane GJ, Marianayagam J. Epidemiology of chronic pain in children and adolescents: a protocol for a systematic review update. BMJ Open. 2021;11(2): PubMed e043675. doi: 10.1136/bmjopen-2020-043675.

[6] Koechlin H, Beeckman M, Meier AH, Locher C, Goubert L, Kossowsky J, Simons LE. Association of parental and adolescent emotion-related factors with adolescent chronic pain behaviors. Pain. 2022 Jul 1;163(7):e888-e898. doi: 10.1097/j.pain.0000000000002508.

[7] Cabanac M. What is emotion?

Behav Processes. 2002; 60(2):69 PubMed -83. doi: 10.1016/s0376-6357(02)00078-5.

[8] Kleinginna, PR, Kleinginna, AM. A categorized list of emotion definitions, with suggestions for a consensual definition. Motiv Emot. 1981; 5, 345-79.

[9] Rosenbloom BN, Katz J. Modeling the transition from acute to chronic postsurgical pain in youth: A narrative review of epidemiologic, perioperative, and psychosocial factors.

Can J Pain. 2022;6(2):166 PubMed -74. doi: 10.1080/24740527.2022.2059754.

[10] Myers K, Winters NC. Ten-year review of rating scales. I: overview of scale functioning, psychometric properties, and

	selection. J Am Acad Child Adolesc Psychiatry. 2002 Feb;41(2):114-22. doi: 10.1097/00004583-200202000-00004. [11] Riley AW (2004) Evidence that school-age children can self-report on their health. Ambul Pediatr. 2004; 4:371–6. doi.org/10.1367/ A03-178R.1 [12] Varni JW, Limbers CA, Burwinkle TM. How young can children reliably and validly self-report their health-related quality of life?: an analysis of 8,591 children across age subgroups with the PedsQL 4.0 Generic Core Scales. Health Qual Life Outcomes. 2007 Jan 3;5:1. doi: 10.1186/1477-7525-5-1. [13] Moon EC, Chambers CT, Larochette AC, Hayton K, Craig KD, McGrath PJ. Sex differences in parent and child pain ratings during an experimental child pain task. Pain Res Manag. 2008;13(3):225-30. doi: 10.1155/2008/457861. [14] Lang TA, Altman DG. Basic statistical reporting for articles published in biomedical journals: the "Statistical Analyses and Methods in the Published Literature" or the SAMPL Guidelines. Int J Nurs Stud. 2015 Jan;52(1):5-9. doi: 10.1016/j.ijnurstu.2014.09.006. Epub 2014 Sep 28. PMID: 25441757 [15] van Smeden M, Moons KG, de Groot JA, Collins GS, Altman DG, Eijkemans MJ, Reitsma JB. Sample size for binary logistic prediction models: Beyond events per variable criteria. Stat Methods Med Res. 2019;28(8):2455-2474. doi: 10.1177/0962280218784726. [16] Den Teuling NGP, Pauws SC, van de Heuvel ER. A comparison of methods for clustering longitudinal data with slowly changing trends. Communications in Statistics - Simulation and Computation 2023; 52, Issue 3 https://doi.org/10.1080/03610918.2020.1861464
--	---

REVIEWER	Gupta, Anju All India Institute of Medical Sciences, Anaesthesiology, pain medicine and critical care
REVIEW RETURNED	03-Jan-2024

GENERAL COMMENTS	Thank you for submitting your valuable work to BMJ. The title does not appropriately reflect the study objectives. The lacuane in existing literature are not identified properly. Other comments are attached in article file.
---

VERSION 1 – AUTHOR RESPONSE

Reviewer 1

This study is the first study to specifically examine the role of emotion-related factors on the child and the parent level in the development of chronic pain after surgery. It has certain academic and practical value. Existing problems:

- Overall, the writing was not standardized enough; The writing logic was common.

We thank the reviewer for this feedback. We believe that the editor's and reviewers' comments have strengthened the writing and the writing logic of the manuscript.

- Lack of necessary description of research results and conclusions in the abstract;

We thank the reviewer for this remark. This is a study protocol, which is why we cannot at this stage include results and conclusions based on these results. The structure of the abstract is based on BMJ Open Formatting Guidelines for study protocols.

- The expression of the research results in the main body was too simple and cannot fully match the research indicators and statistical methods;

We thank the reviewer for this comment. We believe that the changes and additions to the introduction have strengthened the manuscript, now fully matching the research indicators and statistical methods to the expression of existing results. As noted above, the manuscript presents the study protocol, with results not available presently.

- The discussion was not in-depth enough and targeted enough, and the overall level was common.

As per recommendation of the editor, we have deleted the Discussion section, as Discussion/Conclusion sections are not part of journal formatting requirements for protocol articles.

Reviewer 2:

Chronic pain is a serious developmental health concern that can interfere significantly with daily functioning. Children experiencing chronic pain may miss school, withdraw from social activities, and are at risk of developing internalizing symptoms in response to their pain. Given these consequences, issues related to paediatric chronic pain are important to researchers and clinicians to enable the development of effective strategies to ameliorate these problems [1-3].

It is not uncommon for paediatric Chronic Postsurgical Pain (CPSP) research to face challenges in terms of study design and participant age ranges. Many studies focus on specific age groups or narrow age ranges. Due to these shortcomings, there are several gaps in our understanding of age-related differences in the prevalence of pain and the evolutionary trajectory of pain over the lifespan. In order to fully understand the prevalence of pain in children and the risk factors that

predict chronification of acute postoperative pain, it is essential that pain research be more developmentally focused; that is, using broader age intervals and/or conducting more longitudinal studies of pain prevalence will increase understanding of age differences in pain prevalence. [4,5]

We would like to congratulate the authors on their protocol for a longitudinal observational study. This is intended to 1. examine the role of emotional state, emotion variability, emotion regulation, and emotion differentiation at the child and parent levels in the development of CPSP, and 2. to describe individual trajectories related to pain and emotion after surgery.

Emotion-related factors such as those measured in this study can be screened for, and they present as good targets for training modules within the prevention and treatment of pediatric CPSP.

The authors published an earlier study on the association of parental and adolescent emotion-related factors with adolescent chronic pain behaviour [6].

We thank the reviewer for this positive feedback, this is highly appreciated.

However some issues have to be mentioned.

1. The authors wrote
on page 4, lines 7-10:

“So far, most perioperative research has focused on negative emotions.”

And further

“This is the first study to specifically examine the role of emotion-related factors on the child and the parent level in the development of chronic pain after surgery”

a) Although an enormous literature exists on the psychobiology of affect, there is no singular or even preferred definition of emotion. The term is taken for granted in itself and, most often, emotion is defined with reference to a list: anger, disgust, fear, joy, sadness, and surprise. In 1981 Kleinginna and Kleinginna listed 92 different definitions of emotion, compiled from the literature on emotion.

"Emotion" is a broad and multidimensional concept that can be approached from various perspectives depending on the field of study (psychology, neuroscience, sociology, etc.) and the specific context of the research [7,8].

b) A number of child have been evaluated in the prediction of the transition from acute post-operative pain to CPSP. These factors span the domains of pain-related anxiety (pain anxiety, pain catastrophizing, fear of movement) anxiety sensitivity, pain self-efficacy, chronic pain acceptance, symptoms of posttraumatic stress, symptoms of depression, and general anxiety [9].

Moreover, previous studies have shown that parents' vulnerability also affects their child's pain experience and that parents' interpretation of their child's pain may affect children's pain outcomes [9].

Given this evidence from previous research, explain more clearly the novelty of the current study and how the purpose of this study differs from that of previous studies.

VERSION 2 – REVIEW

REVIEWER	Xu, Simao Chengdu Sport University
REVIEW RETURNED	29-Feb-2024

GENERAL COMMENTS	There were some practical consequences and a very clear purpose when designing the study. It is hoped that statistical techniques will be more effectively applied in the particular study procedures based on the real circumstances.
--

REVIEWER	Vanlinthout, Luc KU Leuven University Hospitals Leuven Gasthuisberg Campus Hospital Pharmacy
REVIEW RETURNED	27-Feb-2024

GENERAL COMMENTS	Manuscript ID bmjopen-2023-080174.R1, entitled “Longitudinal Resilience and Risk Factors in Pediatric Postoperative Pain (LORRIS): Protocol for a Prospective Longitudinal Hospital-Based Study.” The paper has been modified and worked upon and comments from the editor and the reviewers have been responded to. The revised manuscript has significantly been improved. Some points, however, deserve further attention.
--

The authors wrote on page 5:
“Children learn the appropriateness of valence, duration, and intensity of emotions and emotion expression by observing their parents and parents’ emotion regulation behaviors (38,39)”.

Children learn a lot by observing the behavior of those around them, especially their parents. But children learn not only by observing. Parents can play an important role in coregulating their children's emotions. Coregulation in this context refers to the process by which a caregiver, usually a parent, provides external regulation or scaffolding for a child to facilitate the development of emotion regulation.

Most research to date has focused on how emotion regulation is influenced by the family and parenting. However, children spend increasing amounts of time in preschool settings, making teachers important agents for children’s development. Furthermore, the preschool setting is characterized by many peer interactions involving conflictual situations that challenge children’s regulatory capacities. Consequently, the preschool setting provides many challenges and a wide range of possibilities for children to obtain and practice these competencies.

Can the authors comment upon this.

The authors wrote on page 7:
“Mental health comorbidities are no reason for exclusion.”

Mental health conditions are more prevalent in children with chronic pain compared to their peers in the general paediatric population. The co-occurrence of chronic pain and various mental health conditions, including posttraumatic stress disorder (PTSD), anxiety and depressive disorders, autism spectrum disorders (ASD), Attention Deficit Hyperactivity Disorder (ADHD), etc. can indeed be attributed to a complex interplay of shared neurobiological, cognitive, and behavioral factors.

Up to this day, only few longitudinal studies have been conducted to prospectively establish the associations of depression, anxiety disorders, PTSD, and personality traits with the incidence or persistence of chronic pain [1].

Can the authors explain how mental health comorbidities will be assessed.

Will the mental health condition be introduced as a covariable in the prediction model?

The model could be confounded by unmeasured depressive symptoms that emerge during the follow-up period. This can blur a true prediction regarding the incidence of chronic pain. How will this be taken into account?

	[1] Rouch I, Strippoli MF, Dorey JM, Ranjbar S, Laurent B, von Gunten A, Preisig M. Psychiatric disorders, personality traits, and childhood traumatic events predicting incidence and persistence of chronic pain: results from the CoLaus PsyCoLaus study. Pain. 2023 Sep 1;164(9):2084-2092. doi: 10.1097/j.pain.0000000000002912.
--	---

REVIEWER	Gupta, Anju All India Institute of Medical Sciences, Anaesthesiology, pain medicine and critical care
REVIEW RETURNED	18-Feb-2024

GENERAL COMMENTS	The concerns have been adequately addressed by the authors
--

VERSION 2 – AUTHOR RESPONSE

Reviewer 1

- There were some practical consequences and a very clear purpose when designing the study. It is hoped that statistical techniques will be more effectively applied in the particular study procedures based on the real circumstances.

We thank the reviewer for raising this important topic. We collaborate with a biostatistician (Prof. Ulrike Held, co-author of the manuscript) throughout the study process to ensure that the collected data is analyzed appropriately.

Reviewer 2

Manuscript ID bmjopen-2023-080174.R1, entitled “Longitudinal Resilience and Risk Factors in Pediatric Postoperative Pain (LORRIS): Protocol for a Prospective Longitudinal Hospital-Based Study.”

The paper has been modified and worked upon and comments from the editor and the reviewers have been responded to. The revised manuscript has significantly been improved.

We appreciate the reviewer’s feedback, and we wanted to express our gratitude for the reviewer’s thorough and helpful comments during the revision process.

Some points, however, deserve further attention.

1. The authors wrote on page 5:

“Children learn the appropriateness of valence, duration, and intensity of emotions and emotion expression by observing their parents and parents’ emotion regulation behaviors (38,39)”.

Children learn a lot by observing the behavior of those around them, especially their parents. But children learn not only by observing. Parents can play an important role in coregulating their children's emotions. Coregulation in this context refers to the process by which a caregiver, usually a parent, provides external regulation or scaffolding for a child to facilitate the development of emotion regulation.

Most research to date has focused on how emotion regulation is influenced by the family and parenting. However, children spend increasing amounts of time in preschool settings, making teachers important agents for children’s development. Furthermore, the preschool setting is characterized by many peer interactions involving conflictual situations that challenge children’s regulatory capacities. Consequently, the preschool setting provides many challenges and a wide range of possibilities for children to obtain and practice these competencies.

- Can the authors comment upon this.

We agree with the reviewer that the preschool setting potentially provides challenges and possibilities regarding the development and practice of emotion regulation strategies. The diary phases in our study allow us to monitor emotion regulation, emotion differentiation, and emotion variability across different settings in participants’ daily lives. School functioning is also included in the baseline surveys (i.e., as a subscale of the Pediatric Quality of Life Survey, PedsQL), which might give us some hints regarding the relationship of emotion regulation and the school setting. Unfortunately, we are unable to contribute to the question in a more specific way, especially as our participants are all over the age of 8 years and hence no longer part of the preschool setting.

2. The authors wrote on page 7:

“Mental health comorbidities are no reason for exclusion.”

Mental health conditions are more prevalent in children with chronic pain compared to their peers in the general paediatric population. The co-occurrence of chronic pain and various mental health conditions, including posttraumatic stress disorder (PTSD), anxiety and depressive disorders, autism spectrum disorders (ASD), Attention Deficit Hyperactivity Disorder (ADHD), etc. can indeed be attributed to a complex interplay of shared neurobiological, cognitive, and behavioral factors.

Up to this day, only few longitudinal studies have been conducted to prospectively establish the associations of depression, anxiety disorders, PTSD, and personality traits with the incidence or persistence of chronic pain [1].

- a. Can the authors explain how mental health comorbidities will be assessed.

We appreciate the reviewer’s comment. Our baseline survey includes questionnaires assessing trauma (Child and Adolescent Trauma Screening, CATS) as well as symptoms of anxiety and depression (Revised Child Anxiety and Depression Scale, RCADS). We measure symptoms of

posttraumatic stress again at our 1stexamine trajectories of mental health symptoms and how they relate to pain trajectories over time.

- b. Will the mental health condition be introduced as a covariable in the prediction model?

We thank the reviewer for this comment. According to published and unpublished data from our collaborator Prof. Jennifer Rabbitts, symptoms of anxiety and depression are associated with the development of chronic postsurgical pain (CPSP). Interestingly, significant differences in symptoms of anxiety and depression seem to be apparent even before the surgery, and continue to be present over 6 months. Therefore, and in consultation with our statistician, symptoms of anxiety and depression were defined as central predictors for the development of CPSP. The corresponding questionnaire is completed by participants at each time point.

- c. The model could be confounded by unmeasured depressive symptoms that emerge during the follow-up period. This can blur a true prediction regarding the incidence of chronic pain. How will this be taken into account?

We agree with the reviewer that unmeasured depressive symptoms can emerge and may then confound the model. We therefore measure symptoms of depression and anxiety by means of the Revised Child Anxiety and Depression Scale (RCADS) at each time point to be able to capture changes in depressive symptoms over time.

[1] Rouch I, Strippoli MF, Dorey JM, Ranjbar S, Laurent B, von Gunten A, Preisig M. Psychiatric disorders, personality traits, and childhood traumatic events predicting incidence and persistence of chronic pain: results from the CoLaus|PsyCoLaus study. *Pain*. 2023 Sep 1;164(9):2084-2092. doi: 10.1097/j.pain.0000000000002912.

Reviewer 3

The concerns have been adequately addressed by the authors.

We thank the reviewer for this feedback and are pleased to have made all adjustments to their satisfaction.

1

VERSION 3 – REVIEW

REVIEWER	Vanlinthout, Luc KU Leuven University Hospitals Leuven Gasthuisberg Campus Hospital Pharmacy
REVIEW RETURNED	12-Mar-2024
GENERAL COMMENTS	I would like to congratulate the authors on this well-designed protocol for a study that is much needed in the field. I am looking forward to the results of their research.